# UniStyle: Unified Style Modeling for Speaking Style Captioning and Stylistic Speech Synthesis

### Xinfa Zhu
Northwestern Polytechnical University
Xi'an, China
xfzhu@mail.nwpu.edu.cn

### Wenjie Tian
Northwestern Polytechnical University
Xi'an, China
twj@mail.nwpu.edu.cn

### Xinsheng Wang
Northwestern Polytechnical University
Xi'an, China
w.xinshawn@gmail.com

### Lei He
Microsoft
Beijing, China
helei@microsoft.com

### Yujia Xiao
Microsoft
Beijing, China
yujiaxiao7@gmail.com

### Xi Wang
Microsoft
Beijing, China
xwang@microsoft.com

### Xu Tan
Microsoft
Beijing, China
xuta@microsoft.com

### Sheng Zhao
Microsoft
Beijing, China
szhao@microsoft.com

### Lei Xie
Northwestern Polytechnical University
Xi'an, China
lxie@nwpu.edu.cn

## Abstract

Understanding the speaking style, such as the emotion of the interlocutor's speech, and responding with speech in an appropriate style is a natural occurrence in human conversations. However, technically, existing research on speech synthesis and speaking style captioning typically proceeds independently. In this work, an innovative framework, referred to as UniStyle, is proposed to incorporate both the capabilities of speaking style captioning and style-controllable speech synthesizing. Specifically, UniStyle consists of a UniConnector and a style prompt-based speech generator. The role of the UniConnector is to bridge the gap between different modalities, namely speech audio and text descriptions. It enables the generation of text descriptions with speech as input and the creation of style representations from text descriptions for speech synthesis with the speech generator. Besides, to overcome the issue of data scarcity, we propose a two-stage and semi-supervised training strategy, which reduces data requirements while boosting performance. Extensive experiments conducted on open-source corpora demonstrate that UniStyle achieves state-of-the-art performance in speaking style captioning and synthesizes expressive speech with various speaker timbres and speaking styles in a zero-shot manner.

## CCS Concepts

• **Computing methodologies** → **Natural language generation**;
• **Human-centered computing** → *Human computer interaction (HCI)*.

## Keywords

Style modeling, speaking style captioning, text-to-speech, data scarcity

**ACM Reference Format:**
Xinfa Zhu, Wenjie Tian, Xinsheng Wang, Lei He, Yujia Xiao, Xi Wang, Xu Tan, Sheng Zhao, and Lei Xie. 2024. UniStyle: Unified Style Modeling for Speaking Style Captioning and Stylistic Speech Synthesis. In *Proceedings of the 32nd ACM International Conference on Multimedia (MM '24), October 28-November 1, 2024, Melbourne, VIC, Australia.* ACM, New York, NY, USA, 10 pages. https://doi.org/10.1145/3664647.3681465

## 1 Introduction

In human speech, speaking style effectively communicates paralinguistic elements, such as emotions, emphasis, and intentions [26]. During interactions, individuals typically understand the speaking style of others and respond with speech in an appropriate style, underscoring the crucial role of speaking style in conversations. In this paper, we pioneer unified modeling for both speaking style understanding and stylistic speech synthesizing.

Earlier work, whether in understanding speaking styles [2, 11] or in synthesizing stylistic speech [1, 28], was conducted based on limited categories, such as several commonly used emotion categories. For instance, Shirian et al. [34] focused on the emotion recognition task and achieved this task with a cross-entropy loss-based classification framework. Correspondingly, in [1, 5], the explicit emotion labels are used as input to control the style of synthesized speech. While these methods can, to some extent, describe or control speaking style, the diversity of speaking styles makes it challenging for such style labels to represent the comprehensive range of real speech styles effectively.

In contrast, natural language exhibits a rich expressive capacity, providing a natural advantage in presenting complex and diverse speech styles. Most recently, several efforts have been conducted towards understanding speech styles using language [42, 43], i.e., generating style descriptions for speech, which task is commonly known as speech style captioning. Yamauchi et al. [43] made a groundbreaking advancement in the field of end-to-end generation

of speaking styles. In their work, an automatic speaking style captioning model, named StyleCap, was proposed to generate speaking captions using a large language model (LLM) with speech representation as input. Compared to conventional classification-based approaches, this speaking style captioning-based method demonstrates notable advantages in the diversity of speaking style understanding, and it offers more dimensions for describing speaking styles, such as variations in pitch and pace.

Similarly, natural language style descriptions have recently been attempted for speech synthesis to control the style of synthesized speech [10, 27, 30, 44]. Beyond single style label-based speech synthesis, PromptTTS [10] proposes to use style descriptions to guide the style expression of synthesized speech successfully across multiple dimensions, such as gender, pitch, pace, volume, and emotion. Recently, PromptTTS 2 [22] introduces a diffusion-based variation network to generate more consistent speech from text descriptions. PromptStyle [27] leverages a cross-modal style encoder to extract style representations from natural language text descriptions and generates high-quality speech through a VITS [19] model. Intuitively, employing natural language descriptions to control speech generation is a convenient and user-friendly way that offers precise control over stylistic attributes.

Although notable progress has been made in speaking style captioning and text description-based speech synthesizing in their respective tasks, it is regrettable that no work has yet unified these two tasks to achieve a human-like ability that encompasses both understanding and expression. As a result, it remains an open question whether these two tasks can mutually benefit and enhance each other's effectiveness despite the intuitive overlap in the information they share.

To bridge this gap and investigate the potential benefits of unifying speaking style captioning and text description-based speech synthesis, we propose a unified model named UniStyle, which is a novel framework designed to have the capability in both speaking style captioning and stylistic speech synthesizing. Specifically, UniStyle comprises a UniConnector and a style prompt-based speech generator. The UniConnector establishes speaking style alignment between speech and text modalities, which extracts style representations and generates style captions for input speech. With the style representations, the style prompt-based speech generator synthesizes expressive speech with a corresponding style expression. Moreover, to address the challenge of data scarcity, we propose a two-stage and semi-supervised training strategy. This strategy optimally leverages both description-labeled and unlabeled data, which exposes UniStyle to large-scale corpora, thus effectively improving the overall performance. Our approach is validated through extensive experiments on the open-source TextrolSpeech [15] and Libriheavy [17] corpora. Results demonstrate UniStyle's superior capability in speaking style captioning and stylistic speech synthesis. Audio samples are available at https://zxf-icpc.github.io/UniStyle/.

The key contributions of our work are summarized as follows:

- We introduce UniStyle, a pioneering framework that establishes a cross-modal speaking style alignment, enabling seamless integration of speaking style captioning and stylistic speech synthesis.

- We propose a novel two-stage and semi-supervised training strategy that reduces data requirements while boosting performance.
- Our approach achieves state-of-the-art performance in speaking style captioning and equips zero-shot TTS with speaking style control, allowing for expressive speech synthesis with various speaker timbres and speaking styles.

## 2 Related Work

### 2.1 Speaking Style Captioning

Speaking styles encompass various aspects, including tone, pitch, rhythm, and emotional expressiveness [15]. Speaking style captioning aims to capture these nuanced features from speech, presenting a significant challenge due to their subtle and context-dependent nature [43]. Conventional approaches focus on classifying and recognizing predefined categories from speech signals [14], typically involving three key components: feature extraction, projection, and classification. Thanks to innovative model architectures, large training corpora, and various loss functions, significant advancements have been achieved in such classification and recognition tasks [2, 9, 11, 29].

However, the complexity and diversity of speaking styles inherent in human speech often render categorization into predefined classes insufficient. Recently, there has been a growing interest in utilizing natural language to describe speaking style in speech. Notably, StyleCap [43] utilizes WavLM [6] for speech feature extraction and trains a mapping network to extract style-related vectors, which are then input into LLaMA [37] to generate detailed speaking style descriptions. Besides, SECap [42] leverages HuBERT [12] for speech feature extraction and employs Q-Former as the Bridge-Net to provide LLaMA with emotion-related speech features. SECap further enhances the captioning capability through diverse instructions and generates precise speech emotion captions.

### 2.2 Stylistic Speech Synthesis

Stylistic speech synthesis addresses the fundamental challenge of TTS, which involves mapping text input to diverse speech outputs with various speaking styles. Mainstream research in this area can be broadly categorized into categorical label-based TTS [1, 20, 28, 48] and reference speech-based TTS [21, 24, 35, 46]. Categorical label-based methods rely on a predefined set of auxiliary categorical labels to represent individual styles. However, the discrete nature of categorical labels often results in over-averaged stylistic expressions, limiting the diversity and naturalness of synthetic speech. In contrast, reference speech-based methods utilize a reference encoder to model various expressive styles from reference speech. While reference speech-based methods offer more flexibility and diversity, the learned style representations lack interpretability, and selecting an appropriate reference speech can be challenging for arbitrary input texts. [45, 47].

A compromise approach has emerged recently, allowing user-friendly control of speaking style in synthetic speech through text descriptions [10, 22, 27, 30, 39, 44]. For instance, PromptTTS [10] proposes to use a style description from five different factors (i.e., gender, pitch, speaking speed, volume, and emotion) to guide the style expression for the generated speech. Additionally, PromptTTS

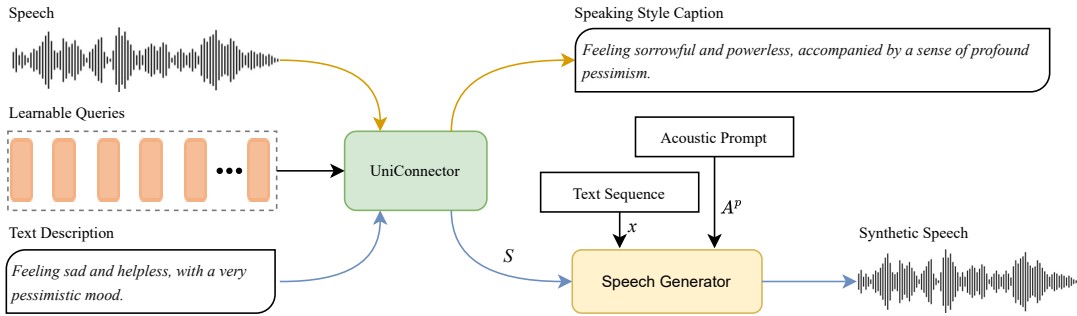

**Figure 1: Overview of the proposed UniStyle.**

2 [22] introduces a diffusion-based variation network, which captures voice variability and produces speech more consistent with text descriptions. InstructTTS [44] synthesizes stylistic speech through a three-stage training approach to capture semantic information from natural language style descriptions. Furthermore, PromptStyle [27] leverages a cross-modal style encoder to extract style representations from natural language text descriptions. These representations are then integrated into an improved VITS [19] model for stylistic speech synthesis. Adopting natural language text descriptions for style control represents a promising direction for the future development of controllable TTS systems, which offer user-friendliness, generalizability, and interpretability.

## 3 Approach

### 3.1 Framework Overview

UniStyle aims to have both the ability to generate style captions from speech input and the capability to synthesize speech with the style described by text descriptions. To achieve this goal, bridging the gap between text description and speech plays a crucial role. As depicted in Figure 1, within the proposed UniStyle framework, a module designed to handle multi-modal input, referred to as UniConnector, assumes this role. To be specific, UniConnector utilizes learnable queries to obtain style information from different modalities, including speech and text, through joint optimization of several objectives. This style information can be directly used by UniConnector to generate textual speaking style captions. Additionally, the style features obtained by UniConnector can serve as the style prompt to assist the speech generator in synthesizing speech with the corresponding style.

In the following sections, we begin by introducing the proposed style prompt-based speech generator, assuming that we have already obtained the style feature $S$ from UniConnector. Then, we will delve into the detailed structure of UniConnector, discussing how it handles speech style captioning, as well as the process of creating style features based on text descriptions.

### 3.2 Style Prompt-based Speech Generator

Inspired by the impressive performance and widespread usage of VALL-E [40], an LM-based TTS model, we adopt a similar structure for the design of the speech generator with an additional style

prompt. This allows for style-controllable speech synthesis via either reference speech or text descriptions.

In the vanilla VALL-E, the speech audio is discretized by a speech codec [7], so that a typical language model can be used to model these discrete tokens based on input texts. Specifically, the speech codec employs a convolutional encoder-decoder model with residual vector quantization (RVQ). The encoder discretizes the speech into acoustic tokens $A_{T*n}$, where $T$ denotes the frame number and $n$ represents the number of codebooks. The speech codec decoder reconstructs the waveform from acoustic tokens. VALL-E regards zero-shot TTS as a conditional codec language modeling task, training neural language models to generate target acoustic tokens $A$ conditioned on a phoneme sequence $x$ and an acoustic prompt $A^p$ with the optimization objective of max $p(A|x, A^p)$. Considering the hierarchical structure of acoustic tokens, VALL-E designs autoregressive (AR) and non-autoregressive (NAR) conditional language models to predict these tokens hierarchically. The AR language model follows a decoder-only transformer architecture and predicts the first-layer acoustic token $A_{:,1}$ conditioned on text sequence $x$ and acoustic prompts $A^p_{:,1}$, which can be formulated as

$$p(A_{:,1}|x, A^p_{:,1}; \theta_{AR}) = \prod_{t=0}^{T} p(A_{t,1}|A_{<t,1}, x, A^p_{:,1}; \theta_{AR}). \quad (1)$$

The NAR language model maintains the same structure as the AR language model, predicting the subsequent layer acoustic tokens $A_{:,2:n}$ based on the first-layer acoustic tokens $A_{:,1}$, text sequence $x$, and acoustic prompts $A^p$, which is expressed as

$$p(A_{:,2:n}|x, A^p; \theta_{NAR}) = \prod_{i=2}^{n} p(A_{:,i}|A_{:,<i}, x, A^p; \theta_{NAR}) \quad (2)$$

However, the use of short, fixed-length acoustic prompts, typically around 3 seconds, poses challenges in conveying a comprehensive speaking style. Different from this method, which relies solely on acoustic prompts, we introduce an additional style prompt to the speech generator. This approach aims to provide more comprehensive style-related information and simultaneously enables control through text descriptions with the support of the proposed UniConnector. Specifically, assuming that we have already obtained the style embedding $S$ from UniConnector, the formulation of modeling

the codec from the first layer can be described as follows:

$$p(A_{:,1}|S, x, A^p_{:,1}; \theta_{AR}) = \prod_{t=0}^{T} p(A_{t,1}|A_{<t,1}, S, x, A^p_{:,1}; \theta_{AR}). \quad (3)$$

For the NAR component, given that the main style-related information is modeled by the AR model, we do not include the style prompt in this part. Instead, we use the same NAR modeling approach as employed in VALL-E.

## 3.3 Multi-modal UniConnector

In the UniStyle framework, the UniConnector is designed to bridge the gap between speech and text modalities, enabling the extraction of style representations and the generation of speaking style captions from text inputs and speech inputs, respectively. Inspired by the Q-former's method [23] for handling cross-modal tasks, the UniConnector employs learnable queries to facilitate various tasks by selectively attending to different information. As shown in Figure 2, the learnable queries interact with input speech using cross-attention. Here, the input speech is presented with the representations extracted by a pre-trained self-supervised learning model called WavLM [6]. The interaction between the learnable queries and text input is achieved by a shared self-attention block, in which the specific attention spans are managed using various attention masks to achieve different purposes, i.e., speech-text matching (STM), speech-text contrastive learning (STC), and speaking style captioning (SSC).

Specifically, STM judges whether the input speech and text descriptions are stylistically consistent by integrating information from both modalities. For this purpose, the learnable queries need to obtain information from the input text and then integrate it with the speech modality via cross-attention. Therefore, the attention mechanism for STM is bidirectional, which implies that the STM mask does not actually employ a masking operation. In practice, the style embedding obtained under the STM mask goes through a binary linear classifier to determine whether the current speech-text pair is matched or not. The corresponding binary classification loss for the optimization is denoted by $\mathcal{L}_{STM}$.

The purpose of STC is to enhance the connectivity between different modal features from a global feature perspective, thereby aligning the style features from text descriptions with the style embeddings of speech within a unified space. This enables the retrieval of the most suitable speech from the database through nearest-neighbor searches based on style text descriptions, thereby facilitating the acquisition of speaking style embeddings $S$ for speech synthesis. To this end, style embedding obtained by the learnable queries should only access speech input rather than the text description. Therefore, the STC mask restricts queries and text embeddings to only attend to themselves, preventing them from accessing the other modality. In practice, we calculate the pairwise similarity between the learnable query-based style embedding and the text-based global feature, typically the embedding of the token [CLS] as that in the BERT [8], and optimize the highest similarity of positive pairs against those of negative pairs, and the corresponding loss is denoted by $\mathcal{L}_{STC}$.

As for the style caption generation, we treat it as an AR generation task so that during the inference stage, text descriptions can

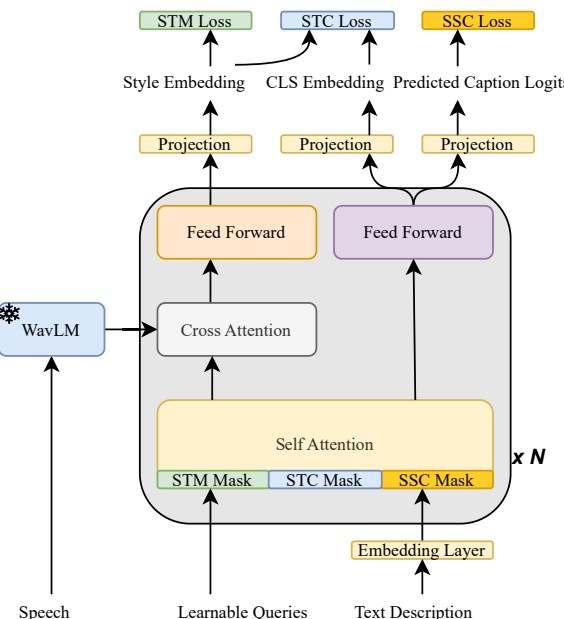

**Figure 2: Model architecture of the UniConnector.**

be progressively generated based on previous information. To this end, following the image-grounded text generation method in [23], a causal self-attention mask, here referred to as SSC mask, is used to control query-text interaction. With the SSC attention mask, the queries can only attend to themselves, while textual token embedding in the current time can only access historical text embeddings but all information from queries. The training loss $\mathcal{L}_{SSC}$ for SSC is a cross-entropy loss for a typical text generation task.

## 3.4 Two-stage and Semi-supervised Training Strategy

To address the challenge of data scarcity, we propose a two-stage and semi-supervised training strategy. We leverage two types of data: a description-labeled corpus $D_l$ containing instances in the format of <text, speech, text description> and an unlabeled corpus $D_u$ comprising instances in the format of <text, speech>.

In the first stage, we pre-train each component of UniStyle separately. The UniConnector is trained on $D_l$ with the following loss function:

$$\mathcal{L}_{align} = \mathcal{L}_{STC} + \mathcal{L}_{STM} + \mathcal{L}_{SSC}, \quad (4)$$

where $\mathcal{L}_{STC}$, $\mathcal{L}_{STM}$, $\mathcal{L}_{SSC}$ represent the loss of speech-text contrastive learning, speech-text matching, and speaking style captioning, respectively. Simultaneously, we pre-trained vanilla VALL-E on $D_u$ as the initiation of the speech generator with the original loss $\mathcal{L}_{LM}$, which calculates the cross-entropy (CE) loss between predicted and ground-truth acoustic tokens. In the second stage, we employ a semi-supervised strategy to fine-tune UniStyle using $D_l$ and $D_u$. The finetuning objective is formulated as follows.

$$\mathcal{L}_{ft} = \sum_{d \in D_l} \mathcal{L}_{align} + \sum_{d \in D_l \cup D_u} \lambda_{LM} \mathcal{L}_{LM}, \quad (5)$$

**Table 1: The corpus used to train each model. '\*' means the corpus is used in the second stage during fine-tuning.**

| Corpus | StyleCap | SECap | PromptStyle | Salle | SC VALL-E | Vec-Tok Speech | UniStyle-P | UniStyle | UniStyle-L |
|---|---|---|---|---|---|---|---|---|---|
| TextrolSpeech [15] | ✓ | ✓ | ✓ | ✓ | ✓ | ✓ | ✓ | ✓ * | ✓ * |
| Libriheavy [17] | | | | | ✓ | ✓ | ✓ | ✓ | ✓ * |

where $\lambda_{LM}$ is the weight of $\mathcal{L}_{LM}$. During the second stage, the parameters of the speech speech generator are frozen.

## 4 Experimental Setup

### 4.1 Dataset

We perform experiments on two open-source corpora. The TextrolSpeech [15] corpus comprises 330 hours of English speech data and 236,220 pairs of style prompts in natural text descriptions, each associated with corresponding speech samples and five style factors, including gender, pitch, speaking speed, volume, and emotion. The Libriheavy [17] corpus, a large-scale ASR corpus, consists of 50,000 hours of read English speech derived from LibriVox. Libriheavy is a labelled version of Librilight [16] with fully formatted text transcripts. We resample all recordings to 16k Hz.

### 4.2 Implement Details

Following Q-Former [23], the query number is set to 32. We initialize the UniConnector with pre-trained weights from the BERT-base model [8], with the cross-attention layers and feed-forward for learnable queries being randomly initialized. We use a pre-trained WavLM-Large model[1] as the frozen speech encoder. Regarding the speech generator, we employ AudioDec [41] as the speech codec and use the libritts v1 model[2] to extract speech tokens. The AR and NAR language models follow the same transformer architecture with 12 layers, 16 attention heads, an embedding dimension of 1024, a feed-forward layer dimension of 4096, and a dropout rate of 0.1. In inference, the length of the acoustic prompt is set to 3 seconds.

In the first stage of training, the UniConnector is pre-trained up to 64k steps on 2 NVIDIA A100 80GB GPUs with a batch size of 160 for each GPU. The speech generator is pre-trained on 8 NVIDIA A6000 48GB GPUs, with a batch size of 6 for each GPU, for 400k steps. We utilize the original optimizers from vanilla Q-Former and VALL-E to optimize each model. In the second stage, we fine-tune UniStyle on 8 NVIDIA A6000 48GB GPUs, with a batch size of 4 per GPU for 300k steps. We modify the data loader to ensure that each mini-batch contains 2 samples with text descriptions and the loss weight $\lambda_{LM}$ is set to 0.5. The models are optimized using the AdamW optimizer, with the learning rate warmed up for the first 50,000 updates to a peak of $1e^{-4}$, followed by linear decay.

During inference, as there are tens of millions of speech utterances in the database, we introduce vector quantization (VQ) to filter out reference speech with expressions of similar style and accelerate the retrieval process. We conduct the k-means algorithm on style embeddings to obtain 8192 clusters. The style embeddings in the center of clusters form a core subset of style expressions. The [CLS] embedding of the input text description retrieves the most suitable style embeddings from the core subset.

### 4.3 Comparison Systems

To assess the performance of UniStyle, we implement the following systems. As those comparison systems have different requirements for corpora, the training corpora for each system are listed in Table 1.

- **StyleCap** [43]: An end-to-end method for generating speaking style prompts from speech. StyleCap utilizes WavLM to extract speech features and trains a mapping network to extract style-related vectors, which are then fed into LLaMA to generate speaking style descriptions.
- **SECap** [42]: A framework that generates high-quality style captions. It uses HuBERT to extract speech features, Q-Former as the Bridge-Net, and LLaMA as the text decoder to produce coherent style captions.
- **PromptStyle** [27]: A VITS model that controls speaking styles through text descriptions. PromptStyle employs a cross-modal style encoder to extract style representations from text descriptions. We replace the speaker ID in PromptStyle with a pre-trained speaker encoder, Resemblyzer[3], to support zero-shot TTS.
- **Salle** [15]: A modified VALL-E model that controls speaking styles through text descriptions. Salle directly prepends text descriptions to the input sequence in the AR language model of VALL-E.
- **SC VALL-E** [18]: A modified VALL-E model that controls speaking styles through reference speech. SC VALL-E incorporates style tokens and designs a style network in the NAR language model of VALL-E.
- **Vec-Tok Speech** [49]: A large-scale speech generation model that controls speaking styles through reference speech. It utilizes a style and speaker prompt to provide style expression for the LM and speaker identity for the codec decoder, respectively.
- **UniStyle-P**: The proposed UniConnector and speech generator of UniStyle, which are pre-trained, respectively.
- **UniStyle**: The proposed framework uses the two-stage training strategy, with fine-tuning conducted on the TextrolSpeech corpus in the second stage.
- **UniStyle-L**: The proposed framework employs the two-stage and semi-supervised training strategy, with fine-tuning conducted on the TextrolSpeech and Libriheavy corpora in the second stage.

### 4.4 Evaluation Metrics

To evaluate the performance of speaking style captioning, we adopt the following metrics commonly used in natural language generation tasks. BLEU@4 [33] measures the n-gram overlap between

---

[1]https://huggingface.co/microsoft/wavlm-large
[2]https://github.com/facebookresearch/AudioDec

[3]https://github.com/resemble-ai/Resemblyzer

generated and annotated captions. METEOR [3] evaluates the overall quality of generated captions by considering precision, recall, and alignment between generated and reference captions. ROUGE-L [25] computes the longest common subsequence between generated and reference captions. CIDEr [38] measures the consensus between generated and reference captions, considering both n-gram overlap and semantic similarity. We adopt sacrebleu[4] to calculate BLEU@4, pycocoevalcap[5] to calculate ROUGE-L and CIDEr, and NTLK[6] to calculate METEOR.

We employ subjective and objective evaluations to assess the performance of stylistic speech synthesis. For subjective evaluation, we conduct two types of human perceptual rating experiments. A total of twenty-one volunteers participate in these experiments. Mean Opinion Score (MOS) is used to evaluate the naturalness of the synthetic speech. Similarity Mean Opinion Score (SMOS) is adopted to evaluate synthetic speech based on style similarity and speaker similarity. The rating criteria are as follows: bad = 1, poor = 2, fair = 3, good = 4, great = 5, with 0.5-point increments. For objective evaluation, we employ an Automatic Speech Recognition (ASR) model to transcribe the generated speech and calculate the Word Error Rate (WER) to assess the robustness of each model. The ASR model[7] is a CTC-based Hubert, pre-trained on Librilight and finetuned on the 960-hour training set of LibriSpeech. In addition, we employ a pre-trained speaker verification model, WavLM-TDCNN[8], to assess speaker similarity (SSIM) between generated samples and acoustic prompt. Moreover, we employ two pitch-related metrics for style similarity: Root Mean Squared Error (RMSE) and Pearson correlation (Corr) [4]. These two metrics are widely applied to evaluate prosody similarity. Since the sequences are not aligned, we perform Dynamic Time Warping to align the sequences before comparison.

## 5 Experimental Results

### 5.1 Automatic Speaking Style Captioning

We preserve a test set comprising 1,000 sample pairs containing speech waveforms and corresponding descriptions to evaluate automatic speaking style captioning. The experimental results, presented in Table 2, highlight the consistent outperformance of the proposed UniStyle family compared to the comparison models. Notably, StyleCap exhibits lower scores in BLEU@4, ROUGE-L, and CIDEr metrics. This performance discrepancy is attributed to the presence of style-irrelevant descriptive words in the output captions of StyleCap. Additionally, StyleCap relies on a global embedding to extract style-related features from speech, which limits its representation ability, particularly for complex speech inputs. In contrast, SECap employs an instruction prompt to constrain LLaMA's output space, resulting in more accurate captions. Moreover, SECap utilizes multiple queries to extract style information from speech, enhancing the representational capacity and overall performance compared to StyleCap.

---

[4]https://github.com/mjpost/sacrebleu

[5]https://pypi.org/project/pycocoevalcap/

[6]https://www.nltk.org/api/nltk.translate.meteor_score.html

[7]https://huggingface.co/facebook/hubert-large-ls960-ft

[8]https://github.com/microsoft/UniSpeech/tree/main/downstreams/speaker_verification

**Table 2: Experimental results on speaking style captioning.**

| Model | BLEU@4 ↑ | METEOR ↑ | ROUGE-L ↑ | CIDEr ↑ |
|---|---|---|---|---|
| StyleCap [43] | 30.8 | 0.161 | 0.151 | 0.051 |
| SECap [42] | 49.7 | 0.160 | 0.168 | 0.160 |
| UniStyle-P | 56.8 | 0.161 | 0.158 | 0.210 |
| UniStyle | **61.1** | 0.208 | 0.196 | 0.340 |
| UniStyle-L | 60.9 | **0.227** | **0.204** | **0.356** |

Among the models within the UniStyle family, UniStyle-P consistently outperforms StyleCap in BLEU@4, ROUGE-L, and CIDEr metrics, and surpasses SECap in BLEU@4, METEOR, and CIDEr metrics. This indicates that the joint optimization of three tasks in the first stage effectively establishes a speaking style alignment between speech and text modalities. Furthermore, UniStyle exhibits superior performance compared to StyleCap, SECap, and UniStyle-P across all metrics by a significant margin, showing UniStyle can acquire more precise style expression from speech. This underscores the joint finetuning with the stylistic speech synthesis task in the second stage benefits the speaking style captioning task, validating the efficacy of the unified modeling of speaking style. Moreover, UniStyle-L performs better than UniStyle in METEOR, ROUGE-L, and CIDEr metrics, highlighting the beneficial impact of leveraging a large-scale unlabeled corpus through the semi-supervised training strategy.

### 5.2 Zero-shot Stylistic Speech Synthesis

We conduct subjective and objective experiments to evaluate each comparison model. Specifically, we preserve 10 transcripts, 20 style prompts (10 reference speech and 10 text descriptions), and 5 acoustic prompts as the test set.

**Subjective evaluation.** As presented in Table 3, UniStyle-L achieves the best performance across all metrics, while most comparison models exhibit unbalanced performance. Notably, PromptStyle demonstrates low speaker similarity due to the limited capability of the pre-trained speaker encoder in zero-shot voice cloning. Salle obtains low scores in naturalness and style similarity, potentially due to text descriptions leaking into the synthetic speech output, resulting in meaningless content in output speech. SC VALL-E shows low style similarity, suggesting that incorporating a style module in the NAR language model of VALL-E has minimal impact on the final synthetic speech, as most apparent style factors, such as speaking speed, are determined by the AR language model of VALL-E. Vec-Tok Speech achieves balanced results; however, UniStyle surpassed Vec-Tok Speech and other comparison models by a remarkable margin. Notably, UniStyle equips the base zero-shot TTS model UniStyle-P with the capability to control speaking style through text descriptions or prompt speech, achieving high style similarity with a slight sacrifice in speaker similarity. Furthermore, UniStyle-L avoids this sacrifice and performs better, indicating that the larger corpus in the second stage contributes to better fine-tuning of UniStyle. These results suggest that the UniStyle family can generate natural speech with various speaking styles and speaker timbre in a zero-shot manner, showcasing its effectiveness in expressive speech synthesis.

**Objective evaluation.** To comprehensively evaluate the performance of zero-shot stylistic speech synthesis, we conduct objective

**Table 3: Experimental results on zero-shot stylistic speech synthesis with 95% confidence interval. Style similarity-T compares synthetic speech to text descriptions, while style similarity-S compares synthetic speech to prompt speech.**

| Model | Naturalness ↑ | Speaker Similarity ↑ | Style Similarity-T ↑ | Style Similarity-S ↑ | WER ↓ | SSIM ↑ | RMSE ↓ | Corr ↑ |
|---|---|---|---|---|---|---|---|---|
| PromptStyle [27] | 3.70 ± 0.06 | 3.23 ± 0.07 | 3.65 ± 0.06 | - | 7.1 | 0.564 | - | - |
| Salle [15] | 2.79 ± 0.15 | 3.71 ± 0.10 | 2.61 ± 0.07 | - | 26.2 | 0.688 | - | - |
| SC VALLE [18] | 3.58 ± 0.08 | 3.74 ± 0.07 | - | 3.44 ± 0.08 | 8.7 | 0.663 | 38.5 | 0.60 |
| Vec-Tok Speech [49] | 3.83 ± 0.07 | 3.70 ± 0.06 | - | 3.78 ± 0.07 | 6.3 | 0.687 | 19.7 | 0.71 |
| UniStyle-P | 3.85 ± 0.10 | 3.87 ± 0.11 | - | - | 6.6 | **0.708** | - | - |
| UniStyle | 3.89 ± 0.12 | 3.81 ± 0.09 | 4.03 ± 0.07 | 3.94 ± 0.08 | 8.2 | 0.692 | 15.4 | 0.78 |
| UniStyle-L | **3.98 ± 0.08** | **3.88 ± 0.07** | **4.11 ± 0.06** | 3.99 ± 0.08 | **6.1** | 0.701 | **12.6** | **0.81** |

**Table 4: Experimental results on the ablation study of different model components and training strategy.**

| Model | Speaking style captioning | | | | Stylistic speech synthesis | | | |
|---|---|---|---|---|---|---|---|---|
| | BLEU@4 ↑ | METEOR ↑ | ROUGE-L ↑ | CIDEr ↑ | WER ↓ | SSIM ↑ | RMSE ↓ | Corr ↑ |
| UniStyle | **61.1** | **0.208** | **0.196** | 0.340 | **8.2** | 0.692 | 15.4 | **0.78** |
| w/o Pre-trained UniConnector | 45.1 | 0.186 | 0.180 | 0.179 | 13.6 | 0.598 | 33.6 | 0.69 |
| w/o UniConnector loss | 1.53 | 0.018 | 0.015 | 0.001 | 8.3 | 0.691 | 50.7 | 0.53 |
| w/ STM | 1.74 | 0.028 | 0.021 | $4e^{-4}$ | 8.4 | 0.688 | 30.2 | 0.67 |
| w/ STC | 0.00 | 0.003 | 0.004 | $5e^{-5}$ | 9.5 | 0.674 | 22.4 | 0.72 |
| w/ SSC | 50.9 | 0.181 | 0.178 | 0.266 | 10.8 | 0.679 | 41.3 | 0.58 |
| w/ Large UniConnector | 52.2 | 0.187 | 0.194 | **0.374** | 8.3 | **0.699** | **14.8** | 0.77 |
| w/ LoRA Speech Generator | 36.0 | 0.183 | 0.184 | 0.276 | 15.6 | 0.504 | 55.4 | 0.51 |

tests to measure WER(%), SSIM, RMSE, and Corr. It is worth noting that PromptStyle and Salle control the speaking style of synthetic speech through text descriptions, not prompt speech; thus, their RMSE and Corr metrics are unavailable. In terms of robustness, UniStyle-L achieves the lowest WER, indicating the good intelligibility of UniStyle-L. Conversely, Salle performs poorly in WER, likely due to content leakage from text descriptions. Regarding speaker similarity, UniStyle-P achieves the highest SSIM, while UniStyle-L obtains a comparable SSIM. Considering the intricate entanglement between speaker timbre and speaking style [31, 36], it is reasonable to exhibit slightly lower speaker similarity when adapting to a new speaking style. Regarding prosody similarity, UniStyle-L achieves better RMSE and Corr than UniStyle and other comparison models, demonstrating the effectiveness of the proposed semi-supervised training strategy, where a larger corpus contributes to better capturing and expressing speaking style. These results confirm the observations from the subjective evaluation, highlighting the superiority of the UniStyle family.

### 5.3 Ablation Study

We conduct an ablation study to assess the effectiveness of each component and training strategy by evaluating their impact on speaking style captioning and stylistic speech synthesis tasks. Additionally, considering the significance of query embeddings in UniStyle, we analyze the influence of different query numbers on UniStyle's performance.

**Ablation study on model components and training strategy**. We perform ablation studies on UniStyle by individually removing the pre-training of the UniConnector in the first stage and the loss of the UniConnector in the second stage. Additionally, we investigate the influence of model size by replacing BERT-base with BERT-large as the initialization of the UniConnector. Furthermore, we explore the effectiveness of the frozen strategy by applying a LoRA finetuning strategy (r=8) [13] to speech generator.

The experimental results are shown in Table 4. Firstly, removing the pre-trained UniConnector leads to a degradation in the performance of speaking style captioning, which indicates that the speaking style alignment established in the pre-training stage plays a vital role in capturing precise style expressions. The performance of stylistic speech synthesis also decreases, with WER and RMSE increasing and SSIM and Corr deteriorating, suggesting that the output query embeddings carried unnecessary information, such as speaker timbre. Secondly, the absence of UniConnector loss in the second stage exhibits a sharp decline in all metrics of speaking style captioning, RMSE and Corr. These results indicate that the model without UniConnector loss fails to accomplish the speaking style captioning task and guide speaking style during speech generation. Furthermore, SSC loss mainly contributes to speaking style captioning. UniStyle with STC loss exhibits better RSME and Corr than with STM or SSC loss, suggesting contrastive learning effectively builds a coarse-grained alignment between two modalities. UniStyle fails in speaking style captioning only with STM loss but generates speech with better WER and SSIM, indicating STM loss is beneficial for model stability. Third, when we take the BERT-large as the initiation of the UniConnector, the overall performance is comparable to that using the BERT-base, indicating the UniConnector initialized with the BERT-base in UniStyle is sufficient for capturing speaking style. Finally, when employing a LoRA finetuning strategy on speech generator in the second stage, the performance of speaking style captioning and stylistic speech synthesis decreases. We find that the output captions become inaccurate, and output speech becomes inconsistent with unstable intelligibility, prosody, and speaker timbre. This observation suggests that finetuning the speech generator, even with few parameters, is not beneficial to UniStyle. Another plausible explanation for this phenomenon could be that the limited dataset is insufficient for effectively finetuning the speech generator.

**Ablation study on the query number**. Given the significance of the query number in UniConnector, we investigate the impact of 8, 16, 32, and 64 queries on the performance of UniStyle. As illustrated in Figure 3, the changing trend of different metrics in speaking style captioning remains consistent when varying the query number, which shows that too-small and too-large query numbers are unsuitable for speaking style captioning. Specifically, when the query number increases, the performance gradually improves because few queries can not model speaking style well due to the limited trainable model parameters. The performance peaks at a query number of 32 and then declines, which we speculate the model with more queries requires a more extensive training corpus.

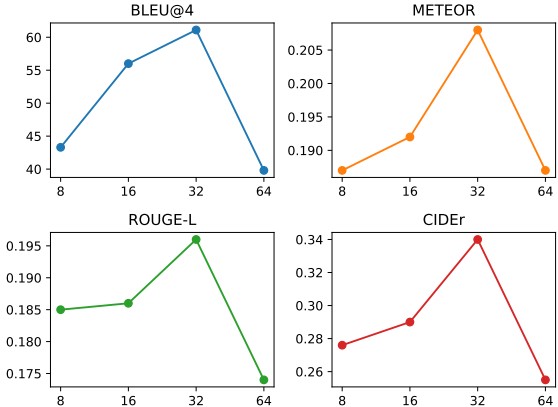

Figure 3: The impact of different query numbers on speaking style captioning.

We further examine the influence of different query numbers on stylistic speech synthesis. As depicted in Figure 4, the tendency differs from that observed in speaking style captioning. Specifically, UniStyle with 16 queries achieves a slightly better WER and Corr but worse SSIM and RMSE than UniStyle with 32 queries. Conversely, when the query number is set to 8 or 64, UniStyle exhibits poor performance, consistent with the findings in speaking style captioning. Overall, the evaluations of speaking style captioning and stylistic speech synthesis emphasize the essential role of the query number in UniStyle, with 32 queries yielding the best performance.

## 6 Discussion and Limitation

We evaluate the effectiveness of UniStyle from the perspective of attribute control. We calculate the classification accuracy for each attribute of synthetic speech. Specifically, we adopt digital signal processing tools[9] to extract pitch, speaking speed, and volume and recognize the category (low/normal/high) according to the statistical range in TextrolSpeech. We use an open-source model[10] to identify gender. For speech emotion recognition, we use another open-source model[11] of emotion2vec [32] and finetune it on the emotion subset of TextrolSpeech.

---

[9]https://github.com/JeremyCCHsu/Python-Wrapper-for-World-Vocoder
[10]https://github.com/karthikbhamidipati/multi-task-speech-classification
[11]https://github.com/ddlBoJack/emotion2vec

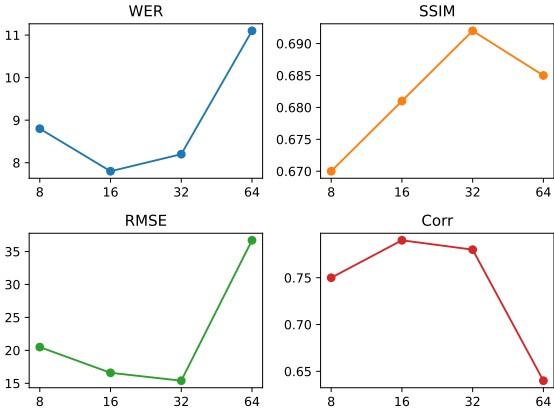

Figure 4: The impact of different query numbers on stylistic speech synthesis.

Table 5: The classification accuracy (%) of synthetic speech of UniStyle and ground-truth speech on the attribute control.

|  | Gender ↑ | Pitch ↑ | Speaking Speed ↑ | Volume ↑ | Emotion ↑ |
|---|---|---|---|---|---|
| GT | 99.6 | 100.0 | 100.0 | 100.0 | 80.6 |
| PromptStyle | **50.2** | 71.3 | 69.4 | 77.4 | 62.8 |
| Salle | 47.1 | 58.2 | 63.5 | 67.1 | 45.9 |
| UniStyle | 45.4 | 86.1 | 91.2 | 90.4 | 74.7 |

Table 5 shows that UniStyle can synthesize speech with reasonable accuracy across all attributes except gender compared to the ground-truth speech. Notably, the accuracy of gender identification is very low. This discrepancy can be attributed to the fact that gender is highly correlated with speaker timbre, which is primarily determined by the acoustic prompt rather than the style prompt during speech generation. In the context of zero-shot TTS, where the synthetic speaker timbre aligns closely with that of the acoustic prompt, such a phenomenon is both expected and reasonable. These findings further underscore the capability of UniStyle to modulate speaking styles under the condition of zero-shot voice cloning.

## 7 Conclusion

This paper proposes UniStyle, a novel framework that unifies speaking style captioning and text description-based speech synthesis. Specifically, we introduce a UniConnector to bridge the gap between text and speech modalities, which extracts style representations and generates style captions from speech inputs. Leveraging the style representations, we introduce a style prompt-based speech generator that synthesizes stylistic speech. Furthermore, We propose a novel two-stage and semi-supervised training strategy that reduces data requirements while boosting performance. Extensive experiments on the open-source corpora demonstrate that UniStyle achieves state-of-the-art performance in speaking style captioning and generates natural speech with various speaking styles and speaker timbre in a zero-shot manner.

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
