# OpenReview forum: "UniStyle: Unified Style Modeling for Speaking Style Captioning and Stylistic Speech Synthesis"
_acmmm.org/ACMMM/2024/Conference — MM2024 Oral_

### Official Review · Reviewer_7eB6 · 2024-05-03

**Rating:** 3
**Confidence:** 3

**Summary:**

In this article, the authors propose a unified framework that integrates Speaking Style Captioning and Stylistic Speech Synthesis into a single model.

**Strengths:**

1. Integrating Speaking Style Captioning with Stylistic Speech Synthesis is both an intuitive and effective idea, which can effectively bridge the gap between these two tasks and make better use of limited datasets.
2. The method proposed in the paper combines the learnable prompt technique with the RVQ technique in audio synthesis, establishing a technical pathway between natural language processing and audio synthesis.
3. The ablation study on the length of learnable queries is very detailed (although it feels like it occupies too much space).
4. The demo page showcases a rich array of experimental results.

**Limitations:**

1. The description of the Style Prompt-based Speech Generator is insufficiently clear.
The meaning of Q in Equation (1) remains unidentified in the paper.

2. The ablation study of the different loss functions in UniConnector
In the author's exposition, UniConnector is a pivotal component that utilizes learnable queries to bridge the gap between the stylistic information in speech and its textual representation. The STM Loss, STC Loss, and SSC Loss are three distinct loss functions, each designed to fine-tune different aspects of the UniConnector. However, the author did not conduct an ablation study that isolates and validates the individual roles or characteristics of these losses, instead treating them collectively. Consequently, the ablation analysis provided is inadequate in thoroughly assessing their respective contributions.

3. In Table 1, in addition to enumerating the datasets employed by each method, it would be beneficial to include a column specifying the task each method is designed to address.

4. In my view, there is room for improvement in the author's figure and table planning abilities to a certain extent. Figure 1 is overly simplistic, failing to exhibit key details of both the speech generator and caption generator. Furthermore, Tables 2 and 5 could have been consolidated. Additionally, the ablation study of the length of learnable queries seems excessively extensive.

**Suitability:**

3

---

### Official Review · Reviewer_H5sL · 2024-05-20

**Rating:** 6
**Confidence:** 4

**Summary:**

This paper proposed UniStyle to incorporate both the capabilities of speaking style captioning and style-controllable speech synthesizing.
The contributions of the paper can be summarized as follows:
-  The authors proposed a framework that establishes a cross-modal speaking style alignment, enabling seamless integration of speaking style captioning and stylistic speech synthesis.
- The authors propose a two-stage and semi-supervised training strategy that reduces data requirements while boosting performance.
- The proposed method achieves state-of-the-art performance in the speaking style captioning task and style-controllable speech synthesizing task.

**Strengths:**

Here are some strengths of this paper:
- Well structured and well presented. The authors clearly describe their methods and demonstrate the effectiveness of the proposed methods using rational approaches.
- Innovative. Introducing a multimodal learning approach to style modeling is a good idea, joint modeling is often more advantageous than direct mapping.
- Adequate experiments and evaluation. The authors have conducted adequate experiments to show that their method achieves leading results by comparing it with some of the current state-of-the-art methods. The authors also further analyze their method through a series of ablation studies. In addition, they provide audio demos to make their results more credible.
- Great practical significance and application value. Speech Style Captioning and Stylistic Speech Synthesis have always been very valuable tasks within the speech field, which can help researchers to better learn style representations or perform data enhancement. It's nice to see such a work that has achieved relatively good results on this task.

**Limitations:**

Here are some limitations of this paper:
- Some of the details are missed.The authors said they use a pre-trained BERT based model to initlize the UniConnector model, but there are 2 Feed Forward modules in the Figure 2, are these modules initlized from the same checkpoint?
- Some results are questionable. I noticed that the ROUGE-L values of the SECap and StyleCap models in the results are quite different from those in the original article, is there a reasonable explanation?

**Suitability:**

3

---

### Official Review · Reviewer_r3Yd · 2024-05-21

**Rating:** 5
**Confidence:** 4

**Summary:**

This paper presents UniStyle, a framework that integrates the capabilities of speaking style captioning and style-controllable speech synthesis. UniStyle addresses the challenge of creating natural and expressive speech by enabling a system to both understand and generate speech with specific speaking styles.
The framework consists of two key components:
UniConnector: This module acts as a bridge between speech audio and textual descriptions. It can generate text descriptions from speech input, and conversely, create style representations from text descriptions for use in speech synthesis.
Style Prompt-based Speech Generator: This component utilizes the style representations generated by UniConnector to synthesize speech with controlled speaking styles.
To address the issue of limited data availability, the authors propose a two-stage semi-supervised training strategy, reducing data requirements while enhancing performance.

**Strengths:**

Integration: UniStyle innovatively combines speaking style captioning and style-controllable speech synthesis, creating a comprehensive framework for generating more natural and expressive speech.
Data Efficiency: The two-stage semi-supervised training strategy addresses the challenge of data scarcity, making the framework more practical and accessible for diverse applications.

**Limitations:**

The paper presents a compelling system innovation with UniStyle, which effectively integrates speaking style captioning and style-controllable speech synthesis. This represents a valuable contribution to the field, particularly in addressing the need for more natural and expressive speech generation.
However, the paper's novelty primarily lies in this system integration rather than groundbreaking advancements in either style captioning or speech synthesis individually. The incremental improvements in style captioning, while notable, could be further highlighted and analyzed.
Furthermore, a crucial aspect missing from the evaluation is the assessment of unseen style caption text generation and annotation. The paper primarily focuses on known styles, but the ability to accurately generate and annotate new, unseen styles is essential for the framework's adaptability and real-world applicability. This aspect should be investigated and discussed by the authors to provide a more comprehensive understanding of UniStyle's capabilities.
Overall, the paper offers a promising system, but additional exploration of unseen style handling and more robust analysis of the framework's strengths and weaknesses in this area would significantly strengthen the paper's contribution.

**Suitability:**

3

---

### Official Review · Reviewer_YoqW · 2024-05-25

**Rating:** 3
**Confidence:** 3

**Summary:**

This paper proposes UniStyle, a novel framework that unifies speaking style captioning and text description-based speech synthesis.
1. The UniConnector aims to bridge the gap between text and speech modalities, which extracts style representations and generates style captions from speech inputs.
2. The style prompt-based speech generator seeks to synthesize stylistic speech.
3. A novel two-stage and semi-supervised training strategy that reduces data requirements while boosting performance.
4. Extensive experiments on the open-source corpora validate the proposed model.

**Strengths:**

1. The paper is well written.
2. The combination of style caption and style speech generation is not very novel. The loss functions and training method is inspiring.
3. The experimental part is also relatively comprehensive.

**Limitations:**

1. In the second training stage, the authors adopt Dl and Du to fine tune the UniStyle, "we employ a semi-supervised strategy to fine-tune UniStyle using Di and Di. The finetuning objective is formulated as follows." However, Du just includes the text and speech, that dont includes the description. I don't know how to use it to train Uniconnector？ Please make it clear.
2. How to fix the value of 𝜆 𝐿𝑀? Maybe more experiments about the hyper-parameter should be better.
3.  The details of MOS evaluation should be added. such as number of volunters, etc.
4. In addition study, there is a lack of various loss function (STM\STC\SSC Losses) with Uniconnector. Which loss is important should be further validate.
5. The Large UniConnector system just replaces the BERT with large one. how about replace the WavLM with traditional acoustic features?
6. In table 6, you just report the results of uniStyle, How about other baselines?

**Suitability:**

2

---

### Meta-Review · Area_Chair_33T1 · 2024-07-02

**Recommendation:** Accept (Oral)
**Confidence:** 4

**Metareview:**

This paper proposes "UniStyle" to incorporate both the capabilities of speaking style captioning and style-controllable speech synthesizing. This means that a textual "style description" can be inferred from speech (in addition to the spoken text), and can be used to modulate speech synthesis for an appropriate response. The paper demonstrates a two-stage semi-supervised learning strategy that seems to achieve good results. The paper fits ACM MM topics and reviews recommend acceptance after author rebuttal.

Reasons to accept:
- The paper is well written and reviewers raise their scores after rebuttal
- The combination of style caption and style speech generation is somewhat novel (VALL-E). The loss functions and training method are inspiring.
- The experimental part is also relatively comprehensive with good ablations, the topic is very timely

Reasons to reject:
- Authors do not mention if and when code, data, and models will be released (they use open source data sets)
- Some results are (still) questionable. Reviewers noticed that the ROUGE-L values of the SECap and StyleCap models in the results are quite different from those in the original article, not sure authors give a reasonable explanation.